# Can Medicinal Plants and Bioactive Compounds Combat Lipid Peroxidation Product 4-HNE-Induced Deleterious Effects?

**DOI:** 10.3390/biom10010146

**Published:** 2020-01-16

**Authors:** Fei-Xuan Wang, Hong-Yan Li, Yun-Qian Li, Ling-Dong Kong

**Affiliations:** 1National Center of Supervision Inspection on Processed Food & Food Additives Quality, Nanjing Institute of Product Quality Inspection, Nanjing 210019, China; 2School of Basic Medical Sciences, Nanjing Medical University, Nanjing 211166, China; hongyanli@njmu.edu.cn (H.-Y.L.); lyq_52245@163.com (Y.-Q.L.); 3School of Life Sciences, Nanjing University, Nanjing 210023, China

**Keywords:** medicinal plants, bioactive compounds, herbs, biological activity, 4-hydroxynonenal, deleterious effects

## Abstract

The toxic reactive aldehyde 4-hydroxynonenal (4-HNE) belongs to the advanced lipid peroxidation end products. Accumulation of 4-HNE and formation of 4-HNE adducts induced by redox imbalance participate in several cytotoxic processes, which contribute to the pathogenesis and progression of oxidative stress-related human disorders. Medicinal plants and bioactive natural compounds are suggested to be attractive sources of potential agents to mitigate oxidative stress, but little is known about the therapeutic potentials especially on combating 4-HNE-induced deleterious effects. Of note, some investigations clarify the attenuation of medicinal plants and bioactive compounds on 4-HNE-induced disturbances, but strong evidence is needed that these plants and compounds serve as potent agents in the prevention and treatment of disorders driven by 4-HNE. Therefore, this review highlights the pharmacological basis of these medicinal plants and bioactive compounds to combat 4-HNE-induced deleterious effects in oxidative stress-related disorders, such as neurotoxicity and neurological disorder, eye damage, cardiovascular injury, liver injury, and energy metabolism disorder. In addition, this review briefly discusses with special attention to the strategies for developing potential therapies by future applications of these medicinal plants and bioactive compounds, which will help biological and pharmacological scientists to explore the new vistas of medicinal plants in combating 4-HNE-induced deleterious effects.

## 1. Introduction

The toxic reactive aldehyde 4-hydroxynonenal (4-HNE) belongs to the advanced lipid peroxidation end products [1]. As the second toxic messenger of free radicals, 4-HNE modifies specific amino acid residues on proteins to form highly cytotoxic reactive 4-HNE adducts via Michael addition or Schiff base reaction or reacts with DNA via epoxidation under oxidative stress conditions. Oxidative stress is one of the main pathogeneses for human disorders [2]. Accumulation of 4-HNE and formation of 4-HNE adducts are induced by mainly reactive oxygen species (ROS) and other free radicals; participate in several cytotoxic processes such as protein dysfunction, apoptosis, and inflammatory injury; and produce deleterious effects [2,3,4,5,6,7,8]. As a result, 4-HNE serves as a biomarker for both lipid peroxidation and oxidative stress [1,2]. Therefore, the inhibition, neutralization, and detoxification of 4-HNE and 4-HNE adducts has become one of the directions of developing potential therapies for oxidative stress-related disorders.

Medicinal plants have been widely used to prevent and treat various kinds of disorders for thousands of years. Their extracts and bioactive compounds have the ability to prevent the cells from oxidative stress injury [9,10,11]. Some of them have been suggested to be attractive sources of potential agents to mitigate oxidative stress and attenuate 4-HNE-induced toxicity. However, researchers pay little attention to reviewing these medicinal plants and natural bioactive compounds on combating 4-HNE-induced deleterious effects in neurotoxicity and neurological disorder, eye damage, cardiovascular injury, liver injury, and energy metabolism disorder [12,13,14,15,16].

Therefore, this review focuses on the therapeutic effects of medicinal plants in treating 4-HNE-related disorders under oxidative stress. An open-ended, English-restricted search of PubMed/MEDLINE database and Web of Science database has been conducted (up to 31 December 2019) using terms such as 4-HNE-induced or driven/induced by 4-HNE, and medicinal plant/herb/phytochemical/natural compounds compound/constituent. This review aims to highlight the pharmacological basis of medicinal plants and bioactive compounds in the attenuation of 4-HNE-induced deleterious effects, as well as the underlying molecular mechanisms. Additionally, this review briefly discusses the strategies for developing potential therapies in future applications of these medicinal plants and compounds, which will benefit biological and pharmacological scientists to explore the new vistas of medicinal plants in combating 4-HNE-induced deleterious effects.

## 2. Neuroprotection

Neurotoxicity exacerbated by oxidative stress is strongly associated with 4-HNE toxicity; its further deterioration cleaves nuclear enzyme poly (ADP-ribose) polymerase (PARP) to affect DNA repair and apoptosis in neurons, leading to neurological disorders [1,17]. Polyphenol extract from red wine offers health benefits and can help to avoid neurodegenerative diseases [18]. This extract inhibits 4-HNE-induced cleavage of PARP and protects against apoptosis in neuronal-like catecholaminergic cells (rat pheochromocytoma, PC12 cells) by reducing intracellular ROS [19]. Polyphenolic flavonoids quercetin and myricetin isolated from red wine extract or *Ginkgo biloba* extract have cytoprotective effects on 4-HNE-induced PC12 cell death [19], further demonstrating the benefits in neurodegeneration [20,21]. Other medicinal plant flavonoids luteolin and apigenin have antioxidant, anti-inflammatory, and neuroprotective effects. These two flavonoid compounds also attenuate 4-HNE-induced PARP-1 and caspase-3 activation as well as cell viability in PC12 cells [22].

The accumulation of 4-HNE up-regulates mitogen-activated protein kinase (MAPK) superfamily, especially C-Jun-N-terminal kinase (JNK), which plays a crucial role in oxidative stress-mediated cellular apoptosis [23,24]. Piceatannol, a bioactive stilbene derivative from medicinal plants *Passiflora edulis* or *Gnetum parvifolium*, restores 4-HNE-induced PARP cleavage and apoptosis regulator Bcl-2 expression, and down-regulates the phosphorylation of JNK (p-JNK) in PC12 cell death and nuclear condensation [25]. Recently, maternal supplementation with piceatannol has been demonstrated to have neuroprotection against neonate brain damage and reverse the sensorimotor deficit and cognitive impairment in rats [26]. Additionally, as a traditional medicinal plant in China and Korea, citri reticulatae viride pericarpium also restores 4-HNE-induced inflammatory injury in PC12 cells via inhibiting the activation of p-JNK and nuclear factor kappa-B (NF-κB) [27]. Mitogen-activated protein kinase kinase 4 (MKK4) as an upstream activator of JNK plays a critical role in response to cellular oxidative stress. Cocoa procyanidin fraction and its major antioxidant compound procyanidin B2 are reported to attenuate 4-HNE-induced nuclear condensation and increase sub-G1 fraction (markers of apoptotic cell death) as well as intracellular ROS accumulation in PC12 cells. This fraction and procyanidin B2 also protect against 4-HNE-induced PC12 cell apoptosis by blocking MKK4 activity [28]. Thus, the modulation of JNK and its upstream activator by medicinal plants and bioactive compounds may have potential therapeutic indications for neurodegenerative diseases.

Nuclear factor (erythroid-derived 2)-like-2 factor (Nrf2) is a ubiquitous master transcription factor that binds to the antioxidant response elements (ARE) and then enhances the expression levels of antioxidant enzymes and cytoprotective genes within DNA in response to oxidative stress [29]. Trans-resveratrol, a natural stilbene constituent isolated from medicinal plant grapevinele, is known to activate Nrf2 signaling pathway. Trans-resveratrol counters the cytotoxic response of 4-HNE consistently and reduces ROS generation and lipid peroxidation in PC-12 cells, showing its improvement of antioxidant defense system. It also restores 4-HNE-induced protein expression changes of mitochondria-mediated apoptosis markers (caspase-3, Bax, and B-cell lymphoma-2 (Bcl-2)) as well as oxidative damage, resulting in the alleviation of apoptotic neurodegeneration in PC12 cells [30]. More importantly, sulforaphane (an isothiocyanate mainly derived from crucifers such as broccoli, cabbages and olives) and carnosic acid (a plastidial catecholic diterpene mainly from *Rosmarinus officinalis*) are two phytochemicals Nrf2/ARE activators, which are found to attenuate 4-HNE-induced inhibition of mitochondrial respiration for complex I; the latter has been found to protect complex II in the isolated cortical mitochondria in young adult male CF-1 mice [14]. Furthermore, sulforaphane and carnosic acid decrease the amount of 4-HNE bound to mitochondrial proteins [14]. Microtubule-associated protein 1 light chain 3 alpha (LC3) is a marker of autophagosome, which is required for clearing dead cells. Study has shown that polyphenolic flavonoids luteolin and apigenin mitigate LC3 conversion and ROS production, activate Nrf2 signaling, and inhibit cytotoxicity in 4-HNE-exposed PC12 cells [22]. Therefore, the above phytochemicals have the capability of Nrf2 signaling induction to prevent the mitotoxicity of 4-HNE and provide neuroprotection.

Hyperactivity of NADPH oxidases (NOX) induced by 4-HNE has a crucial role in ROS overproduction and apoptosis; thus, NOX-derived ROS is a central mechanism in the development of neurodegeneration and neuroinflammation [31]. Kaempferol, a natural flavonoid presented in many medicinal plants, has been proved as a potential neuroprotective agent. It suppresses 4-HNE-mediated p-JNK and apoptosis in PC12 cells. Interestingly, kaempferol directly binds p47(phox), a cytosolic subunit of NOX, and effectively blocks 4-HNE-induced NOX activation in PC12 cells, exerting its potential neuroprotection against NOX-mediated neurodegeneration [13]. Scutellarin, a bioactive flavone isolated from *Scutellaria baicalensis*, has anti-inflammatory, anti-oxidative, anti-apoptotic, and anti-neurotoxic properties. More recently, a molecular docking study showed that scutellarin selectively binds to NOX2 with high affinity. Substantially, scutellarin down-regulates NOX2 expression and reduces 4-HNE and ROS levels in astrocytes subjected to ischemia/reperfusion and in rats with focal cerebral ischemia [32]. These results indicate that scutellarin is a neuroprotective flavone against ischemic injury.

Alzheimer’s disease (AD) is a progressive and irreversible neurodegenerative disorder. High levels of 4-HNE are detected in the AD brain with the induction of neuronal apoptosis and oxidative stress, indicating that 4-HNE may play a role in AD subjects and/or animal models of age-related neurodegenerative disorders [33,34]. Acetylcholine esterase (AChE), cyclooxygenase-2 (COX-2), and matrix metalloproteinase-8 (MMP-8) are important target proteins implicated in AD pathogenesis. Phytocompounds albiziasaponin-A, iso-orientin, and salvadorin have antioxidant and anticholinesterase activities. The results from molecular docking studies show that there are strong interactions of albiziasaponin-A, iso-orientin, and salvadorin with AChE, COX2, and MMP8, respectively. Hence, these three bioactive compounds reduce serum 4-HNE levels in colchicine-induced AD model of rats, suggesting that albiziasaponin-A, iso-orientin, and salvadorin may be potential neuroprotective agents [35].

Glutathione *S*-transferase (GST) is a critical enzyme that participates in the detoxification of 4-HNE and has potential to prevent degenerative cellular processes [36]. Naringenin, a flavonoid found in grapefruit juice, is reported to improve learning and memory in a rat model of AD [37]. In an intracerebroventricular-streptozotocin-induced AD-type model of rats with cognitive impairment, naringenin increases GST, glutathione peroxidase (GSH-Px), glutathione reductase (GR), superoxide dismutase (SOD), and catalase (CAT) to detoxify 4-HNE in the hippocampus [38]. Collapsin response mediator protein-2 (CRMP-2) is hyperphosphorylated in AD. More recently, using computational tools with structural and dynamic analyses, naringenin and naringenin-7-O-glucuronide have been found to selectively bind CRMP-2 and then reduce p-CRMP-2, being the therapeutic activity of novel CRMP-2 inhibitor [39,40].

In senescence-accelerated mouse prone 8 (SAMP8) model of mice with age-related cognitive decline and AD, rosemary extract as well as spearmint extract (containing carnosic acid and rosmarinic acid) are reported to reduce 4-HNE levels in brain cortex and protein carbonyls in brain hippocampus, resulting in the improvement of learning and memory [41]. More importantly, rosmarinic acid increases antioxidant enzyme activities (such as SOD, CAT, and GSH-Px) and glutathione levels and then reduces 4-HNE in amyloid beta 42-induced echoic memory decline in rats with oxidative stress and cholinergic impairment, possibly contributing to the improvement of neural network dynamics of auditory processes [42]. Dark tea prevents and treats age-related degenerative diseases. l-theanine, 3,3′-azanediylbis (4-hydroxybenzoic acid) and one of 8-C N-ethyl-2-pyrrolidinone substituted flavan-3-ols were recently isolated for in silico characterization of microbial metabolites extracted from dark tea. Oral administration of these three compounds indeed inhibits the formation of 4-HNE and ubiquitinated protein aggregates, regulates Aβ metabolic pathway, increases endogenous antioxidant capacity, and reduces neuronal apoptosis rate in SAMP8 mice, showing the protection of SAMP8 neurons [43].

Parkinson’s disease (PD) is an age-related neurodegenerative disorder that severely affects quality of life. Chrysin, a flavone found in *Passiflora caerulea*, *P. incarnata*, and *Oroxylum indicum*, improves antioxidant defense system to protect against oxidative stress by increasing GSH to reduce 4-HNE. Consistently, chrysin improves behavioral and cognitive ability in a 6-hydroxydopamine-induced mouse model of PD [44]. Quercetin also markedly improves the motor balance and coordination in 1-methyl-4-phenyl-1, 2, 3, 6-tetrahydropyridine-induced PD of mice. Moreover, quercetin significantly decreases 4-HNE immunoreactivity in striatum of mouse brains and increases GPx, SOD, AchE, and dopamine contents, showing the antiparkinsonian property [45].

## 3. Prevention of Eye Damage

The main visual ocular structures are the cornea and conjunctiva, lens, vitreum, and retina. Any structure dysfunction can cause visual problems and even blindness. Overproduction of ROS leads to the increased level of 4-HNE and induces lipid oxidative damage and inflammation in the cornea and conjunctiva [46,47], whereas 4-HNE accumulation causes the death of lens epithelia cells to induce lens opacification and cataract [48] as well as retinal pigment epithelium (RPE) cell death [49,50]. In the ocular surface of mouse model with dry eye disease, antioxidant extracts from *Schizonepeta tenuifolia*, *Angelica dahurica*, *Rehmannia glutinosa*, and *Cassia tora* reduce the number of 4-HNE-positive cells to protect from lipid peroxidation-related membrane damage. These antioxidant extracts also inhibit extracellular ROS production and decrease IL-1β, IL-6, tumor necrosis factor-α (TNF-α), and interferon-γ (IFN-γ) levels of the ocular surface, resulting in the improvement of clinical signs [51].

Aldose reductase (AR, an NADPH-dependent oxidoreductase) hyperactivity in the lens plays an important role in the pathogenesis of oxidative stress-mediated diabetic cataract [52]. Quercetin possesses therapeutic effect in the management and treatment of diabetic cataract. Quercetin and its glycoside derivative rutin, or *G. biloba* extract, remarkably inhibit AR activity, stimulate GSH production, and decrease the levels of 4-HNE, lipid peroxidation malondialdehyde (MDA), and advanced glycation end-products in the lenses of streptozotocin-induced diabetic cataract rats, delaying the progression of lens opacification [53]. Curcumin, a diketone constituent extracted from *Curcuma longa*, is a wonderful natural antioxidant and anti-cataract agent. This bioactive compound enhances the formation of GST isoenzymes to detoxify 4-HNE by aiding in the conjugation of 4-HNE to glutathione, showing its prevention against cataract [48].

Proliferative diabetic retinopathy, a leading cause of blindness, is also promoted by 4-HNE. Berberine, an isoquinoline alkaloid from a medicinal plant rhizoma of *Coptidis chinesis*, has favorable effects on glucose and lipid metabolism in animal experimental and human clinical studies [54]. In confluent human retinal Müller cells exposed to 4-HNE, cell death is partially attenuated by berberine pretreatment. Berberine also restores 4-HNE-induced autophagy in this cell model, showing its potential to prevent diabetic retinopathy [55].

In fact, 4-HNE can destabilize actin and induce cytoskeletal damage, resulting in retinopathy and age-related macular degeneration [56]. Cyanidin-3-glucoside is a major anthocyanin in mulberry or *Lonicera caerulea,* and its supplement has potential to prevent eye diseases. In human adult retinal pigmented epithelial (ARPE-19) cells, cyanidin-3-glucoside protects against 4-HNE-induced cell apoptosis, inflammatory damage, and angiogenesis [12,57]. Quercetin also increases viability and decreases inflammation and cytotoxicity in 4-HNE-exposed ARPE-19 cells [58]. Recently, solid dispersion of quercetin has been reported to decrease retinal pigment epithelium sediments and Bruch’s membrane thickness in Nrf2 wild-type (WT) mice with dry age-related macular degeneration. This solid dispersion reduces ROS and MDA contents and increases SOD, GSH-PX, and CAT activities in serum and retinal tissues of Nrf2 WT mice. Solid dispersion of quercetin also up-regulates Nrf2 mRNA expression and enhances its nuclear translocation, as well as Nrf2 target gene hemeoxygenase-1(HO-1) in retinal tissues of Nrf2 WT mice [59]. These observations suggest that quercetin may alleviate oxidative injury to prevent dry age-related macular degeneration by enhancing Nrf2 activation. Medicinal plants marigold or grape seed (containing macular pigments lutein and zeaxanthin) are reported to have anti-oxidative activity and prevent 4-HNE adduct formation, actin solubility, and lipofuscin accumulation as well as age-related cone and rod photoreceptor dysfunction in β5(-/-) mice with cytoskeletal damage in aging RPE cells [56]. Of note, 4-HNE release and oxidative damage are also induced by irradiation exposure, causing retinopathy. Cyanidin-3-glucoside and quercetin decrease 4-HNE release in rod outer segments incubated with all-trans-retinal to generate bisretinoid under irradiation [60], whereas lutein and zeaxanthin isomers have recently been demonstrated to protect against light-induced retinopathy by reducing oxidative and endoplasmic reticulum stress in BALB/cJ mice [61]. Photodegradation of N-retinylidene-N-retinylethanolamine (A2E) is known to release reactive carbonyls. Medicinal plant compounds cyanidin-3-glucoside, quercetin, ferulic acid, and chlorogenic acid diminish cellular ROS and protect GSH from the reaction with photooxidized A2E in A2E accumulated-RPE cells irradiated with short-wavelength light [60].

This is especially important in exploring the possible molecular mechanisms by which bioactive compounds prevent 4-HNE-related retinopathy. NOD-like receptor protein 3 (NLRP3) inflammasome activation in the eyes is associated with the pathogenesis of age-related macular degeneration in RPE cells [62]. Cyanidin-3-glucoside has been found to inhibit NLRP3 inflammasome activation by reducing NLRP3, caspase-1, IL-1β, and IL-18 levels in 4-HNE-exposed ARPE-19 cells. This inhibitory mechanism may be mediated by regulating JNK-c-Jun/activator protein 1 (AP-1) pathway, further demonstrating the potential of cyanidin-3-glucoside to prevent retinal degenerative diseases [63,64]. Quercetin improves cell membrane integrity and mitochondrial function and decreases IL-6, IL-8, and monocyte chemotactic protein 1 (MCP-1) production, presumably by regulating MAPK/NF-κB signaling pathway in ARPE-19 cells stimulated by inflammatory cytokine [65].

## 4. Protection against Cardiovascular Injury

The normal redox status can be altered by 4-HNE adducts, which cause cardiomyocyte damage and vascular dysfunction. Thus, 4-HNE may play a key role in the progression of cardiovascular diseases [5,66,67]. *Thymbra capitata* is a Mediterranean culinary herb. Its essential oil prevents 4-HNE-induced cell death and avoids mitochondrial membrane potential loss and ROS generation in primary cultures of neonatal rat cardiomyocytes [15]. Olive (*Olea europaea*) leaf, a popular traditional herbal medicine, has a cardioprotective function. The ethanolic and methanolic extracts of *O. europaea* are reported to inhibit 4-HNE-induced apoptosis, ROS production, viability impairment, mitochondrial dysfunction, and pro-apoptotic activation in rat cardiomyocyte cell line (H9c2 cells). These extracts also reduce 4-HNE-induced phosphorylation of stress-activated transcription factors in H9c2 cells [68]. Furthermore, phenolic compounds oleuropein, hydroxytyrosol, and quercetin derived from olive leaf are found to prevent against 4-HNE-induced cardiomyocyte carbonyl stress and toxicity and eventually regulate the cellular redox status [68]. Other medicinal plant constituents such as epigallocatechin-3-gallate, sulforaphane, and puerarin are reported to protect against cardiomyopathy. These compounds also reduce 4-HNE and lipid peroxidation to prevent oxidative stress in animal models of cardiovascular injury [69,70,71].

Aldehyde dehydrogenase 2 (ALDH2) mediates detoxification of toxic aldehydes, being necessary and sufficient to confer cardioprotection. Alpha-lipoic acid isolated from spinach and broccoli acts as a free radical scavenger. This bioactive compound activates myocardial ALDH2 and reduces 4-HNE and MDA levels in a Langendorff model of rat ischemia/reperfusion [72]. A recent report shows that α-lipoic acid prominently down-regulates the expression of 4-HNE and NOX subunit p67^phox^ to inhibit oxidative stress in fistula-created deterioration of cardiac function rats [73], further demonstrating the cardioprotective effect. Flavonoid glycoside baicalin, extracted from *Scutellaria baicalensis* root, protects against hypoxia/reoxygenation-induced H9c2 cell injury. Baicalin also enhances ALDH activity to reduce of 4-HNE and MAD levels and increases SOD activity to suppress ROS production in H9c2 cells, showing its cardioprotection against hypoxia/reoxygenation-induced cardiomyocyte injury [74].

Atherosclerotic lesion is associated with the accumulation of reactive aldehydes derived from oxidized lipids. Accumulation of 4-HNE becomes an important risk factor that contributes to the atherogenicity of oxidized low-density lipoprotein (ox-LDL) and the development of atherosclerosis [75]. *G. biloba leaf* extract is reported to reduce ox-LDL and attenuate 4-HNE-induced production of matrix metalloproteinase-1 (MMP-1), probably through suppressing the activation of tyrosine-phosphorylated form of platelet-derived growth factor receptor beta in human coronary smooth muscle cells [76]. Medicinal plant *Opuntia cladodes* powder containing phenolic acid and flavonoids also reduces ox-LDL and 4-HNE toxicity in normal (Apc +/+) and preneoplastic (Apc min/+) immortalized epithelial colon cells [77]. Moreover, *O. cladodes* significantly reduces the formation of atherosclerotic lesions and the accumulation of 4-HNE adducts in the vascular wall of apoE-KO mice [78]. These observations demonstrate that *O. cladodes* may have therapeutic potential in 4-HNE-related cardiovascular diseases.

## 5. Protection against Liver Injury

There is evidence that the increased level of 4-HNE is correlated with the pathogenesis of liver injury [2,79]. Ginseng is a well-known medicinal plant. Ginseng extracts exert protection against liver injury via multiple pathways. For example, the fine root extract of ginseng with ginsenosides profiles has been found to attenuate 4-HNE-induced DNA damage in HepG2 cells. Methanolic extract of the main root also shows a decrease of DNA damage in this cell model [80]. Water extract of *P. ginseng*, rich in ginsenosides Rg1 and Rb1, significantly reduces total ROS generation and especially down-regulates 4-HNE signals in radiation-induced steatohepatitis of mice [81]. Red ginseng or black ginseng is a type of the repeated steam-processed *P. ginseng*. Red ginseng extract improves chronic alcohol-induced histopathological changes of liver in mice. This extract also inhibits oxidative stress and lipid peroxidation by reducing the formation of 4-HNE and the number of 4-HNE-positive cells and protects hepatocytes from inflammation and necrosis by activating AMP-activated protein kinase (AMPK)/Sirt1 pathway [82]. Recently, black ginseng extract has been reported to maintain the cellular redox status by restoring NOX and GSH level change and alleviate oxidative stress by reducing intracellular ROS production and lipid peroxide 4-HNE signals, resulting in the protection of hydrogen peroxide-induced oxidative damage in AML-12 cells [83]. American ginseng berry, the ripe fruit of *Panax quinquefolius*, contains a variety of protopanaxadiol ginsenosides and protopanaxatriol ginsenosides. This ripe fruit extract inhibits alanine aminotransferase and aspartate transaminase activity and mitigates oxidative stress and lipid peroxidation by reinforcing SOD activity, suppressing GSH depletion, and reducing 4-HNE formation in acetaminophen-induced liver injury of mice [84].

Several other bioactive compounds of medicinal plants also mitigate oxidative stress, improve antioxidant function, and reduce 4-HNE, protecting against liver injury. Tanshinone II-A is a diterpene quinone constituent of *Salvia miltiorrhiza*. This bioactive compound restores 4-HNE-induced hepatocyte damage in normal liver tissue NCTC 1469 cells. Tanshinone II-A also up-regulates peroxisome proliferator-activated receptor α (PPARα) expression and scavenges 4-HNE in this hepatocyte model [85]. Fisetin, one of the most popular polyphenols in fruits and vegetables, exhibits senotherapeutic activity in treating aged mice. It reduces liver oxidative stress damage by decreasing 4-HNE adducts and increasing the ratio of intracellular oxidized glutathione [86,87]. *p*-Coumaric acid, rich in fruits, vegetables, and plants, protects hepatocytes against oxidative stress and lipid peroxidation efficiently by reducing high levels of ROS and formation of 4-HNE protein adducts. Consistently, this active compound attenuates acetaminophen-induced hepatotoxicity, apoptosis, and inflammation in mice by suppressing the translocation of apoptosis-inducing factor and MAPK signaling and decreasing NF-κB activity [88]. Chlorogenic acid is an active compound in *Ligustrum lucidum* fruit that protects hepatocyte mitochondria from oxidative stress mediated by the activation of AMPK in carbon tetrachloride-induced acute liver injury of mice [89]. Water extract of *L. lucidum* inhibits ROS generation and increases intracellular GSH to reduce 4-HNE levels in this animal model, demonstrating that *L. lucidum* inhibits oxidative stress to prevent liver injury via AMPK activation [89].

Of note, high 4-HNE levels cause rapid cell death associated with the depletion of sulfhydryl groups and inhibition of key metabolic enzymes. Aldo-keto reductase family 7 member A2 (AKR7A2) is the most abundant anthracycline metabolizing enzyme, which has been found to protect against 4-HNE toxicity. AKR7A2 over-expression rescues the effect of Nrf2-knockdown on 4-HNE-induced cytotoxicity [90]. The active coumarin compound 7-Hydroxycoumain is presented in many medicinal plants, such as the seeds of *Clausena lansium*, *Parasenecio forrestii*, *Artemisia argyi*, and *Pharbitis purpurea*. This phytochemical acts as an AKR7A2 inducer and protects against 4-HNE-induced HepG2 cell damage via AKR7A2 induction. Thus, the inducible AKR7A2 has provided a new therapeutic target to treat oxidative stress-related chronic liver disease [90]. Additionally, the effects of bioactive compounds on AR activity have been reported. Plant triterpenoid saponins soyasaponin Ba (V), soyasaponin Bb, soyasaponin Bd (sandosaponin A), soyasaponin αg, 3-O-[R-L-rhamnopyranosyl (1→2)-α-D-glucopyranosyl (1→2)-α-D-glucuronopyranosyl]olean-12-en-22-oxo-3α,-24-diol, and soyasaponin βg isolated from methanolic extract of *Phaseolus vulgaris* seeds (Zolfino bean) have the ability to inhibit highly purified human recombinant aldose reductase (hAKR1B1), as well as hAKR1B1-dependent reduction of 4-HNE [91]. This work suggests that the reduction of 4-HNE by metabolic enzymes is an important determinant of the prevention of oxidative stress-driven liver injury.

## 6. Improvement of Energy Metabolism Disorder

Oxidative stress induced by 4-HNE also causes energy metabolism disorder that leads to tissue injury [92,93,94]. A harmful effect is exerted on adipocytes by 4-HNE, and 4-HNE also promotes insulin resistance [92,95,96]. In fact, 4-HNE induction and its lipolytic response are observed in insulin resistance patients with obesity and type 2 diabetes mellitus [92]. Insulin signaling pathway plays an important role in mediating energy metabolism. Tocotrienol-rich fraction in olive oil is reported to protect against oxidative stress, inflammation, and apoptosis, prevent the increased levels of 4-HNE and protein carbonyls from hyperglycemia-induced muscle damage, and effectively reduce insulin resistance in type 2 diabetic mice, via regulating the AMPK/Sirt1/peroxisome proliferator-activated receptor γ coactivator 1 (PGC1) pathway [97]. Carnosic acid decreases 4-HNE-mediated free fatty acid release and activates Tyr632 phosphorylation of insulin receptor substrate-1 (p-IRS-1^Tyr632^) and protein kinase B (p-Akt), inhibits as well as p-IRS-1^Ser307^ in insulin signaling pathway in 3T3-L1 adipocytes. This rosemary diterpene compound also suppresses 4-HNE-induced phosphorylation of protein kinase A (p-PKA) and hormone-sensitive lipase (HSL) and attenuates 4-HNE-induced down-regulation of p-AMPK and acetyl-CoA carboxylase, demonstrating the alleviation of insulin resistance driven by 4-HNE [16].

The improvement of antioxidant defense system in treating energy metabolism disorders is also important. GST can protect against 4-HNE-induced insulin resistance [36]. Carnosic acid up-regulates GST expression to reduce 4-HNE-conjugated proteins, being consistent with its attenuation of 4-HNE-induced lipolytic response and insulin resistance [16]. *Morus alba* (mulberry), a popular ornamental plant, has several well-documented beneficial effects to prevent and treat metabolic diseases [98]. The root bark extract of *M. alba*, rich in Diels–Alder type adducts cudraflavone B and other flavonoids, markedly protects against lipid peroxidation-induced pancreatic injury by increasing GSH and reducing 4-HNE in rats [99]. Mulberry leaf contains a variety of flavonoids to control obesity [100]. Recent study shows that mulberry leaf aqueous extract significantly reduces plasma levels of 4-HNE adducts and enhances the cellular antioxidant defense system to inhibit oxidative stress via normalizing Nrf2 activation in high fat diet-fed mice [101]. Mulberry fruits, a well-known source of resveratrol, prevent diabetic complications by reducing lipid peroxidation and 4-HNE accumulation in diabetic pregnancy [102]. Consistently, mulberry bioactive compound trans-resveratrol mitigates oxidative stress by reducing lipid peroxidation and inhibiting 4-HNE expression and DNA damage in streptozotocin-induced type 1 diabetes of rats [103]. In addition, the inhibition of NOX plays a key role in the mitigation of oxidative stress and prevention of insulin resistance, steatosis, and muscle damage [104,105]. NOX inhibitor apocynin, isolated from *Picrorhiza kurroa*, has been found to attenuate insulin resistance in skeletal muscle of mice with metabolic syndrome. Another phytochemical, (−)-epicatechin, found in green tea and cocoa, also down-regulates NOX1/NOX4, reduces 4-HNE adducts from ileum cells, and mitigates high fat diet-induced insulin resistance and steatosis in male C57BL/6J mice by preventing oxidative stress [106]. Cyanidin and delphinidin are the most common aglycon forms of anthocyanins and also have NOX inhibitory activity to attenuate high fat-induced steatosis and control the adverse effects by reducing 4-HNE-protein adducts in mice [107].

Another risk factor of obesity and type 2 diabetes is increased following exposure to fine particular matter (PM_2.5_) [108,109,110]. Research has shown 4-Hydroxytyrosol to be beneficial for metabolic and cardiovascular disorders. It also reduces liver and serum 4-HNE levels and alleviates PM_2.5_-induced, adiposity, and insulin resistance in adult female C57BL/6j mice by mitigating oxidative stress as well as restraining NF-κB activation and gut microbiota [111]. Additionally, there is evidence that 4-hydroxytyrosol significantly reduces 4-HNE expression in red blood cells from hyperlipemic patients and protects red blood cells against 4-HNE toxicity [112].

## 7. Amelioration of Other Disorders

Medicinal plants and bioactive compounds also combat 4-HNE-induced deleterious effects and ameliorate the clinical signs in other disorders. *Aloe vera* is a medicinal herb that contains numerous bioactive components. Aqueous extract of *A. vera* has an antioxidant capacity on endogenous ROS production and 4-HNE-protein adducts induced by 4-HNE in human cervical cancer (HeLa), human microvascular endothelial cells (HMEC), human keratinocytes (HaCat), and human osteosarcoma (HOS) cell cultures [113]. Triterpene saponin aescin isolated from *Aesculus hippocastanum* restricts lipid peroxidation to reduce cytotoxic 4-HNE levels and keeps more neurons and myelin sheaths alive in the spinal cord injury of rats, which may be mediated by enhancing HO-1 expression and inhibiting NF-κB activation [114,115]. Natural anti-oxidation compounds capsaicin (chili peppers), curcumin (turmeric), and polyphenols as well as *G. biloba* and *Polypodium leucotomos* are reported to inhibit 4-HNE-induced oxidative stress in human melanocytes, resulting in the inhibition of cell apoptosis [116]. Rutin effectively decreases 4-HNE and increases GSH-Px and GSH; thus, it prevents against UV-induced skin fibroblast membrane disruption via the regulation of antioxidant system [117]. Ascorbic acid has a similar effect in reducing 4-HNE levels in human skin fibroblasts exposed to UV radiation and hydrogen peroxide [118].

Additionally, 4-HNE is reported to disrupt gap junction-mediated intercellular communication in the lateral wall structures of the cochlea [119]. Some medicinal plant constituents such as astragaloside IV, baicalein, catalpol, curcumin, kaempferol, luteolin, quercetin, resveratrol, and rosmarinic acid are reported to prevent hair cell death [120]. Astragaloside IV (the major constituent of *Astragalus membranaceus*) significantly down-regulates 4-HNE expression in guinea pig cochlea exposed to impulse noise, being consistent with the attenuation of impulse noise-induced trauma [121,122]. Rosmarinic acid enhances the endogenous antioxidant defense and decreases 4-HNE expression by the regulation of the Nrf2/HO-1 signaling pathway, resulting in the attenuation of noise-induced hearing loss and injury in rat cochlea [123]. Therefore, these natural compounds may be the effective therapy to ameliorate 4-HNE-driven hearing loss.

During food digestion, the absorption of highly toxic 4-HNE from the gastrointestinal tract into the blood system may contribute to the development of multiple oxidative stress-related disorders [124]. Polyphenol-rich beverages such as Japanese Sencha green tea and grape juice promote the stability of polyunsaturated fatty acids to oxidation and inhibit 4-HNE formation during the simulated digestion of linseed oil emulsions in the intestinal phase [125]. Algae (microalgae *Schizochytrium* sp.), the division of lower plants, also has pharmaceutical applications. Algae oil, rich in phenolic compounds β-carotene and tocopherols, is reported to prevent lipid peroxidation and inhibit 4-HNE formation in digestion during both in vitro gastric and duodenal digestions [126]. (−)-epicatechin also restores fat diet-induced increase of intestinal permeability, down-regulation of ileal tight junction proteins, as well as reduction of endotoxemia, being consistent with the improvement of insulin resistance. More importantly, (−)-epicatechin and apocynin are found to suppress NOX1/NOX4 overexpression, 4-HNE adducts and monolayer permeabilization by regulating redox sensitive signals in TNFα-exposed Caco-2 cells, further demonstrating the protection against the high fat diet-induced increased intestinal permeability. These results are consistent with the prevention of steatosis and insulin resistance under the treatment of (−)-epicatechin and apocynin [106]. The edible amaranth plants from around the world have strong antioxidant activity [127]. Clinical study shows that the supplementation of amaranth seed oil in addition to standard anti-helicobacter pylori treatment significantly decreases the accumulation of 4-HNE-histidine adducts in gastric mucosa and increases heart rate variability in duodenal peptic ulcer patients. Therefore, the standard treatment of duodenal peptic ulcer requires additional therapeutic approaches by using amaranth seed oil [128].

Energy metabolism disorders induced by 4-HNE may alter longevity by orchestrating the development of a biological phenotype. Plant fruits rich in high levels of antioxidants can promote longevity and health span. For example, the supplementation with 4% nectarine has been found to extend life and increase fecundity. Nectarine reduces 4-HNE-protein adduct in wild-type females fed with high-fat diet. Moreover, nectarine ameliorates aging-related death and reduces oxidative damage in female sod1-mutant flies [129]. These results demonstrate that the attenuation of 4-HNE-induced energy metabolism disorders by medicinal plants and bioactive compounds may promote longevity.

## 8. The Strategy for Developing Potential Therapy

As a toxic end-product of lipid peroxidation, 4-HNE is an important mediator in physiological adaptive reaction and signal transport, as well as in the pathogenesis of multiple oxidative stress-related disorders. Medicinal plants and bioactive compounds are found to enhance metabolism, detoxification or clearance of 4-HNE by regulating activities of endogenous enzymes (such as AR, GST, and AKRs), which may provide the effective strategies for combating 4-HNE-driven deleterious effects. Because of differential metabolisms of 4-HNE observed in liver, lung, and brain of rodents [130], further study is warranted to determine whether these medicinal plants and bioactive compounds have the capacity to mediate oxidative, reductive, and conjugative pathways to metabolize 4-HNE and 4-HNE adducts in oxidative stress-triggered injury of these tissues.

The Nrf2/ARE pathway is suggested to mediate the adaptive induction of antioxidant and detoxifying enzymes including GST and AKR1C1, resulting in the metabolism of 4-HNE [131]. The capability of the Nrf2/ARE pathway induction may constitute a pleiotropic cytoprotective defense to prevent 4-HNE toxicity. Ongoing studies will determine the therapeutic effects of Nrf2/ARE activators derived from medicinal plants and bioactive compounds to mitigate 4-HNE cytotoxicity-induced pathophysiology of disorders or diseases.

Of note, PPARα modulates several biological processes that are perturbed in energy homeostasis [132]. A recent report shows that PPARα is unable to block β-catenin transcriptional activity induced by a constitutively active mutant of lipoprotein receptor-related protein 6 (LRP6). In fact, 4-HNE-induced ROS production enhances LRP6 stability; this event is inhibited by PPARα overexpression [133]. It is known that anti-obesity hormone leptin signaling as well as insulin signaling mediate the regulation of PPARα. Insulin resistance is improved by 4-HNE [96], which also selectively inhibits leptin signaling, possibly promoting the pathogenesis of leptin resistance in obesity [134]. Some medicinal plants and natural compounds acting on PPARα, leptin, or insulin signaling may be responsible for the therapeutic effects on 4-HNE-induced energy metabolism disorders [85].

However, several important issues related to 4-HNE-driven molecular events in diseases, or the relevance of medicinal plants and bioactive compounds in redox homeostasis, still need more studies and new comprehensive approaches. In this regard, preclinical studies and clinical intervention trials are required, which should include the use of accurate analytical techniques, such as the determination of 4-HNE and 4-HNE adducts by immunohistochemistry and enzyme-linked immunosorbent assay, as well as matrix-assisted laser desorption/ionization–tandem time of flight (MALDI–TOF/TOF) mass spectrometry and liquid chromatography–tandem mass spectrometry (LC–MS/MS) [135,136,137,138].

## 9. Conclusions and Future Perspectives

As a biomarker of lipid peroxidation and oxidative stress, cytotoxic 4-HNE contributes to the progression of oxidative stress-related disorders. Some medicinal plants and bioactive compounds have been demonstrated to reduce 4-HNE levels, detoxify 4-HNE, and inhibit 4-HNE adduct formation through the up-regulation of the antioxidant enzymes and suppression of 4-HNE-mediated downstream signals. More experiments have proved that these medicinal plants and bioactive compounds combat 4-HNE-induced deleterious effects in oxidative stress-related neurological disorder, eye damage, cardiovascular injury, liver injury, energy metabolism disorder, and other disorders (Table 1). The understanding of the molecular basis for the function of medicinal plants and bioactive compounds would be useful to facilitate the selection of 4-HNE metabolism for future intervention investigation and health claim support and develop new treatment for 4-HNE-induced deleterious effects in these disorders. Moreover, the identification and analysis of 4-HNE and 4-HNE adducts are necessary to further support the association between medicinal plants and oxidative stress-related disorders in clinical trials.

## Figures and Tables

**Table 1 biomolecules-10-00146-t001:** Medicinal plants and bioactive compounds in the attenuation of 4-HNE-induced deleterious effects.

Medicinal Plants and Bioactive Compounds	Action/Mechanism	4-HNE-Induced Model	References
Neuroprotection			
Polyphenol extract (red wine)	Inhibition of cleavage of PARP, reduction of ROS, protection against apoptosis	PC12 cells with apoptosis	[19]
Quercetin Myricetin (red wine extract or *G. biloba*)	Cytoprotective effects	PC12 cell death	[19,20,21]
Luteolin Apigenin (plant flavones)	Attenuation of cell death, caspase-3 and PARP-1 activation, mitigation of LC3 conversion and ROS production, activation of Nrf2 signaling	PC12 cells with cell viability	[22]
Piceatannol (*P. edulis* or *G. parvifolium*)	Cytoprotective effect, restoration of PARP cleavage and Bcl-2 expression, down-regulation of p-JNK	PC12 cell death and nuclear condensation	[25]
Citri reticulatae viride pericarpium	Anti-inflammation	PC12 cells with inflammatory injury	[27]
Cocoa procyanidin fraction Procyanidin B2	Attenuation of nuclear condensation, apoptotic cell death and ROS accumulation, blockade of MKK4 activity	PC12 cell death and nuclear condensation	[28]
Trans-resveratrol (grapevinele)	Countering the cytotoxic response, attenuation of apoptotic neurodegeneration	PC12 cells with cytotoxicity	[30]
Sulforaphane (crucifers such as broccoli, cabbages and olives) Carnosic acid (*R. officinalis*)	Increase of mitochondrial respiration Nrf2/ARE induction preventing against mitotoxic effect	Young adult male CF-1 mice Isolated cortical mitochondria with inhibition of mitochondrial respiration	[14]
Kaempferol (flavonoid in many medicinal plants)	Suppression of apoptosis and p-JNK, inhibition of NOX activation	PC12 cells with apoptosis Neuron-like cells with NOX-mediated neurodegeneration	[13]
Prevention of eye damage			
Berberine (*C. chinesis*)	Restoration of autophagy, inhibition of diabetic retinopathy	Confluent human retinal Müller cells with cell death	[55]
Cyanidin-3-glucoside (plant fruits mulberry or *L. caerulea*)	Reduction of apoptosis ratio, inflammation and angiogenesis	ARPE-19 cells with apoptosis, inflammatory damage and angiogenesis	[12,63]
Cyanidin-3-glucoside	Inhibition of NLRP3 inflammasome activation Regulation of JNK-c-Jun/AP-1 pathway	ARPE-19 cells with inflammation	[64]
Quercetin	Anti-inflammation, improvement of cell membrane integrity and mitochondrial function, decrease of IL-6, IL-8 and MCP-1 production, regulation of MAPK pathway	ARPE-19 cells with cytotoxicity	[58,65]
Protection of cardiovascular injury			
Oil of *T. capitata*	Prevention of cell death, mitochondrial membrane potential loss and ROS generation	Primary cultures of neonatal rat cardiomyocytes with cell death	[15]
Ethanolic and methanolic extracts of olive leaf (*O. europaea*)	Inhibition of apoptosis, ROS production, viability impairment, mitochondrial dysfunction and pro-apoptotic activation, reduction of phosphorylation of stress-activated transcription factors	Rat cardiomyocytes with cell death	[68]
Oleuropein Hydroxytyrosol Quercetin (olive leaf)	Prevention of carbonyl stress and toxicity, regulation of cellular redox status	Rat cardiomyocytes with cell death	[68]
*G. biloba* leaf extract	Reduction of ox-LDL, attenuation of MMP-1 production, inhibition of the tyrosine-phosphorylated form of platelet-derived growth factor receptor beta activation	Human coronary smooth muscle cells with injury	[76]
*O. cladodes* powder (containing phenolic acid and flavonoids)	Protection against toxicity	Normal (Apc +/+) and preneoplastic (Apc min/+) immortalized epithelial colon cells with toxicity	[77]
Protection against liver injury			
Fine root extract of ginseng with ginsenosides profiles Methanolic extract of the main root	Inhibitory capacity against DNA damage	HepG2 cells with DNA damage	[80]
Tanshinone II-A (*S. miltiorrhiza*)	Alleviation of hepatocyte damage up-regulation of PPARα, and scavenging 4-HNE	NCTC 1469 cells with damage	[85]
7- Hydroxycoumain (*C. lansium*, *P. forrestii*, *A. argyi*, and *P. purpurea*)	Hepatoprotection via AKR7A2 induction	HepG2 cells with cytotoxicity	[90]
Improvement of energy metabolism disorder			
Carnosic acid (*R. officinalis*)	Reduction of free fatty acid release, activation of p^Tyr632^ IRS-1 and p-Akt, p^Ser307^IRS-1, suppression of the PKA/HSL pathway activation, decrease of p-AMPK and acetyl-CoA carboxylase, alleviation of insulin resistance	3T3-L1 adipocytes with insulin signaling impairment	[16]
Carnosic acid	Attenuation of free fatty acid release, up-regulation of GST, reduction of 4-HNE-conjugated proteins attenuation of the lipolytic response	Human subcutaneous adipocytes with lipolysis	[16]
4-Hydroxytyrosol (olive leaf)	Protection of red blood cells with oxidative damage	Hyperlipemic patients	[112]
Repair of other disorders			
*A. vera*	Antioxidant capacity for the reduction of ROS and 4-HNE-protein adducts	HeLa, HMEC, HaCat, and HOS cells with over-production of ROS and -HNE-protein adducts	[113]
Capsaicin (chili peppers) Curcumin (turmeric) Polyphenols *G. biloba* extract *P. leucotomos* extract	Inhibition of oxidative stress and cell apoptosis	Human melanocytes with oxidative stress and apoptosis	[116]

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
