# Peer review of "Can Medicinal Plants and Bioactive Compounds Combat Lipid Peroxidation Product 4-HNE-Induced Deleterious Effects?"

_biomolecules, 2020, doi:10.3390/biom10010146_

Round 1

Reviewer 1 Report

Dear Authors,

Thank you for submitting your review manuscript to Biomolecules Journal.

This is well organized and well written, but there are some points to be added for better understanding.

1) The methodology part is missing how you collected all the literature and information regarding this article.

2) The figure containing overview of this article including the mode of action for 4-HNE related to the compound class which are mentioned in the table might be helpful to readers

3) In some point, the size and style of letters are not uniformed. Please check overall the manuscript.

4) In Table 1, the plant source of the compound (which you mentioned in the manuscript) can be added.

Best,

Author Response

Response to Reviewer 1 Comments

Point 1: Thank you for submitting your review manuscript to Biomolecules Journal. This is well organized and well written, but there are some points to be added for better understanding.

The methodology part is missing how you collected all the literature and information regarding this article.

Response 1: Thank you for your suggestions. The open-ended, English-restricted search of PubMed/MEDLINE database and Web of Science database have been conducted (up 31 December 2019), using the terms such as 4-HNE-induced or driven/induced by 4-HNE, and medicinal plant /herb /phytochemical /compound /constituent, which was presented in the Introduction of the revised manuscript.

Point 2: The figure containing overview of this article including the mode of action for 4-HNE related to the compound class which are mentioned in the table might be helpful to readers

Response 2: As suggested, we have revised Graphical Abstract, which contained overview of this review including the mode of action for 4-HNE related to the compound class.

Point 3: In some point, the size and style of letters are not uniformed. Please check overall the manuscript.

Response 3: We have checked overall the manuscript (the size and style of letters).

Point 4: In Table 1, the plant source of the compound (which you mentioned in the manuscript) can be added.

Response 4: As suggested, we have added the plant source of the compound in the Table 1.

Reviewer 2 Report

The article contains interesting information regarding the free radical properties of diverse phytochemicals. However, the information is not presented well and lacks more detailed explanation of certain terms.

You state that NRF2 is ¨phytochemical¨, instead of a transcription factor.

Many terms are not adequately explained, which makes it more difficult to follow for the reader, unless he or she is a specialist in the field. There is inconsistency in writing of the some scientiifc names, as most are written in italics in some sections, but not in others.

The references are sometimes underlined in bold, which indicates that an adequate revision was not undertaken by the authors.

Finally, there is a need for a complete revision of the English grammar and syntax.

Author Response

Response to Reviewer 2 Comments

Point 1: The article contains interesting information regarding the free radical properties of diverse phytochemicals. However, the information is not presented well and lacks more detailed explanation of certain terms.

Response 1: Thank you for your suggestions. More detailed explanation of certain terms have been added in the revised manuscript.

Point 2: You state that NRF2 is ¨phytochemical¨, instead of a transcription factor.

Response 2: Sorry, we have checked the manuscript, not found this statement “NRF2 is ¨phytochemical¨.

Point 3: Many terms are not adequately explained, which makes it more difficult to follow for the reader, unless he or she is a specialist in the field. There is inconsistency in writing of the some scientiifc names, as most are written in italics in some sections, but not in others.

Response 3: Thank you for your comments and suggestions. We have re-written and revised this manuscript. We have added the information about the detailed explanation of certain terms. We also corrected writing of the some scientific names in the manuscript. The Latin names of medicinal plants are in italics.

Point 4: The references are sometimes underlined in bold, which indicates that an adequate revision was not undertaken by the authors.

Response 4: Thank you for your suggestion. We have checked the references in the revised manuscript.

Point 5: Finally, there is a need for a complete revision of the English grammar and syntax.

Response 5: We have carefully checked and corrected the English grammatical errors and syntax errors throughout the manuscript.

Round 2

Reviewer 2 Report

The paper shows improvement, but still needs to be revised in order to enhance the fluidity of the information, versus snippets of data placed together.

Additionally, the proper English syntax and grammar (especially regarding articles and tenses) needs to be revised.

Author Response

Response to Reviewer 2 Comments

Point 1: The paper shows improvement, but still needs to be revised in order to enhance the fluidity of the information, versus snippets of data placed together.

Response 1: Thank you for your kind help. As suggested, we re-revised and re-wrote this manuscript to enhance of the information. 

Point 2: Additionally, the proper English syntax and grammar (especially regarding articles and tenses) needs to be revised.

Response 2: Many thanks. As suggested, the manuscript undergo thorough correction for English syntax, grammar grammar and spelling errors. This manuscript has been proofread by a native speaker.